# Multiscale Analysis and Validation of Effective Drug Combinations Targeting Driver KRAS Mutations in Non-Small Cell Lung Cancer

**DOI:** 10.3390/ijms24020997

**Published:** 2023-01-05

**Authors:** Liana Bruggemann, Zackary Falls, William Mangione, Stanley A. Schwartz, Sebastiano Battaglia, Ravikumar Aalinkeel, Supriya D. Mahajan, Ram Samudrala

**Affiliations:** 1Department of Biomedical Informatics, University at Buffalo, Buffalo, NY 14260, USA; 2Department of Medicine, University at Buffalo, Buffalo, NY 14260, USA; 3Roswell Park Cancer Institute, Buffalo, NY 14203, USA

**Keywords:** multiscale drug discovery, multitargeting, translational bioinformatics, KRAS, G12C, NSCLC, tyrosine kinase inhibitors, KRAS inhibitors, drug synergy

## Abstract

Pharmacogenomics is a rapidly growing field with the goal of providing personalized care to every patient. Previously, we developed the Computational Analysis of Novel Drug Opportunities (CANDO) platform for multiscale therapeutic discovery to screen optimal compounds for any indication/disease by performing analytics on their interactions using large protein libraries. We implemented a comprehensive precision medicine drug discovery pipeline within the CANDO platform to determine which drugs are most likely to be effective against mutant phenotypes of non-small cell lung cancer (NSCLC) based on the supposition that drugs with similar interaction profiles (or signatures) will have similar behavior and therefore show synergistic effects. CANDO predicted that osimertinib, an EGFR inhibitor, is most likely to synergize with four KRAS inhibitors.Validation studies with cellular toxicity assays confirmed that osimertinib in combination with ARS-1620, a KRAS G12C inhibitor, and BAY-293, a pan-KRAS inhibitor, showed a synergistic effect on decreasing cellular proliferation by acting on mutant KRAS. Gene expression studies revealed that MAPK expression is strongly correlated with decreased cellular proliferation following treatment with KRAS inhibitor BAY-293, but not treatment with ARS-1620 or osimertinib. These results indicate that our precision medicine pipeline may be used to identify compounds capable of synergizing with inhibitors of KRAS G12C, and to assess their likelihood of becoming drugs by understanding their behavior at the proteomic/interactomic scales.

## 1. Introduction

Lung cancer is the most lethal cancer in the world, with non-small cell lung cancer (NSCLC) accounting for about 80% of cases [1]. The proto-oncogene Kirsten rat sarcoma homolog *KRAS* is a member of the RAS family of oncogenes and is the most commonly mutated across all cancers [2]. Somatic nonsynonymous mutations in the *KRAS* gene cause single amino acid substitutions in the small GTPase KRAS it encodes [2]. These mutations confer resistance to many available therapies for NSCLC, generally lead to a more aggressive disease progression, and lack viable treatment options [3]. Despite decades of research attempting to directly inhibit KRAS, little progress has been made due to its high binding affinity with GTP at the active site, and lack of other available binding pockets [3].

A specific KRAS protein mutation, glycine to cysteine at residue 12 (G12C), occurs in about 13% of NSCLC patients [3]. A new class of irreversible covalent inhibitors has been shown to bind the cysteine 12 in the KRAS G12C mutant, which alters its conformation and decreases the ability of the protein to exchange GDP for GTP, thereby increasing affinity for its inactive GDP-bound form [3]. This leads to a decrease in cellular proliferation, as when KRAS is in the GDP-bound form it is unable to trigger downstream proteins involved in cellular proliferation and survival such as Raf-MEK-ERK and PI3K-AKT-mTOR (Figure 1) [1]. Variations of KRAS G12C inhibitors are now being developed by Amgen, Mirati, J&J and Wellspring Biosciences, Sanofi, and X-Chem [3]. Determining which of these inhibitors is most effective for patients with NSCLC and other cancers with KRAS G12C mutations, and whether combining these inhibitors with other drugs leads to better treatment outcomes, are key emerging questions.

Wellspring Biosciences first developed ARS-853, a KRAS G12C specific covalent inhibitor, based on pioneering work by Shokat and colleagues identifying KRAS-G12C specific allosteric inhibitors [3,4]. This compound then evolved to ARS-1620 through a new interaction with the histidine at position 95 (H95) which increased potency [3]. ARS-1620 was further optimized by exploiting a surface groove on KRAS G12C, made possible by an alternative orientation of H95, giving rise to the compound AMG-510 aka sotorasib [3]. Similarly, Mirati Therapeutics used a structure-based approach to design MRTX1257, followed by MRTX849 aka adagrasib, which was granted FDA approval shortly after sotorasib [3].

In addition to KRAS G12C specific inhibitors, another class of drug is being developed to target the interaction between KRAS and son of sevenless homolog 1 (SOS1) [5]. As part of the KRAS scaffold, along with Src homology region 2 domain-containing phosphatase 2 (SHP2), SOS1 aids in the transformation of the GTP-bound active form of KRAS to the inactive GDP-bound form. Since the active form of KRAS leads to increased cellular proliferation, and KRAS mutations such as G12C increase the time the protein remains active, targeting this interaction is another key avenue for intervention [5]. BAY-293 has emerged as a KRAS-SOS1 interaction targeting inhibitor that has shown effectiveness in targeting both wild type and mutant KRAS-SOS1 interactions, and has demonstrated synergy with ARS-853 in targeting KRAS G12C in vitro [5].

### 1.1. Upstream and Downstream Inhibitors for Treatment of KRAS-Mutant NSCLC

Recent studies have shown that second generation epidermal growth factor receptor (EGFR) inhibitors may also be used to treat KRAS-mutant cancers such as NSCLC [6]. EGFR is located upstream of the KRAS pathway and commonly contains difficult to treat mutations occurring in NSCLC, many conferring drug resistance to known inhibitors [6]. First generation inhibitors such as erlotinib and gefitinib are only capable of targeting EGFR and EGFR mutations; however, KRAS mutations allow for increased expression of ErbB2 and ErbB4 and therefore are not susceptible to EGFR inhibition alone [6]. Historically, EGFR inhibitors have not been effective against KRAS-mutant cancers; however, second generation inhibitors have shown promise in treating both KRAS- and EGFR-mutant tumors by targeting all members of the ErbB family of proteins (EGFR/ERbB1, HER2/ErbB2, HER3/ErbB3, HER4/ErbB4) [6].

As first, second, and third generation EGFR/ErbB inhibitors have distinct mechanisms of action, it is necessary to determine which of these may be shared with KRAS G12C inhibitors such as ARS-1620 [6]. Erlotinib and gefitinib are first generation EGFR inhibitors, capable of binding wild type and mutant EGFR proteins [7]. Unfortunately, patients develop resistance to first generation EGFR inhibitors over time by acquiring the threonine to methionine mutation at position 790 (T790M) [8]. In response to these mechanisms of resistance, second generation inhibitors afatinib and dacomitinib were developed specifically to simultaneously and irreversibly inhibit multiple ErbB receptors [8]. Although afatinib and dacomitinib showed greater effectiveness than gefitinib and erlotinib, both are still unable to inhibit cancer caused by resistance to the T790M mutation [7]. To address this, osimertinib was developed to bind to EGFR T790M, while sparing wild type EGFR, and has shown success in treatment as both a first and second line option for EGFR-mutant NSCLC [8].

Targeting signaling cascades downstream of KRAS such as RAF-MEK-ERK kinases with MEK inhibitors has also been used to treat KRAS-mutant NSCLC [9]. Unfortunately, this approach yielded limited success due to resistance driven by increased activation of ErbB2 [6,9]. However, treating KRAS G12C cells with second generation EGFR inhibitors in combination with KRAS G12C inhibitors may mediate resistance mechanisms [10]. Specifically, resistance to ARS-1620 was mediated by treatment with the second generation EGFR inhibitor afatinib [10]. Further analysis revealed that this resistance is modulated by EGFR and aurora kinase A (AURK) signaling [10]. The relationship between AURK and EGFR to these resistance mechanisms, and whether third generation EGFR inhibitors such as osimertinib may show similar effects is yet to be determined. Recently, overexpression of HER2 specifically has been shown to cause resistance to KRAS G12C inhibitors such as sotorasib/AMG-510, which can in turn be mediated with SHP2 inhibitors [11].

Osimertinib has been shown to exhibit increased ability to target ErbB2/HER2, when compared to second generation EGFR inhibitors in mouse models of HER2 [12]. Since osimertinib has demonstrated the capacity to target ErbB2 as well as EGFR mutations, the combination of osimertinib and ARS-1620 may also alleviate ErbB2 driven resistance mechanisms. Finally, osimertinib has a more favorable toxicity profile than first generation and second generation EGFR inhibitors, making it a potentially stronger candidate for drug combination [8]. Figure 1 illustrates the key pathways both upstream and downstream that are relevant for targeting KRAS G12C.

### 1.2. Computational Approaches for Generating Drug Candidates

Computational studies provide valuable insights into drug behaviors and functions, which is useful for predicting new drug candidates, particularly for oncogenic mutations lacking treatment options [10,13,14,15,16,17,18,19]. Growing research has demonstrated that drugs likely target multiple protein receptors beyond their approved use, which lends itself to the concept of drug repurposing where the same drug approved for one indication/disease is used for another [18,19]. Computational analysis of drug repurposing at the proteomic/interactomic scale may yield insights into specific pathways involved in complex indications/diseases such as NSCLC, as well as other cancers. This is particularly relevant as drug discovery utilizing traditional methods, such as high throughput screens, has slowed within the pharmaceutical industry [20,21]. Drug repurposing aids in the speed of drug development by finding new uses for existing drugs with known safety profiles [20]. The Computational Analysis of Novel Drug Opportunities (CANDO) platform for multiscale therapeutic discovery, repurposing, and design has been developed to overcome current issues in drug discovery, such as the simultaneous optimization of efficacy and safety, by using a holistic multiscale approach to characterize drug/compound behaviors and functions [21,22,23,24,25,26,27,28,29,30,31,32,33,34].

A typical pipeline in CANDO calculates interactions between every compound and every protein from large libraries to generate a compound-protein interaction matrix where each compound (each row in the matrix) is described by a set of interactions to all the proteins (each column in the matrix) in a proteome. These compound-proteome interaction signatures are compared and analyzed to assess behavioral/functional similarity of drugs/compounds known to be effective for an indication/disease. The goal of our study was to apply CANDO towards predicting novel drugs/compounds capable of synergizing with known KRAS inhibitors, and validate these findings using human cancer cell lines. To this end, we compared compound-proteome interaction signatures to compounds that are known to be effective against particular mutants of KRAS, which were then used to generate novel predictions of compounds and synergies that may be effective against KRAS-mutant driven cancers and validated (Figure 2). Our predictions using CANDO indicated that EGFR inhibitors synergize with KRAS G12C inhibitors, and that osimertinib had the highest likelihood of synergizing relative to other EGFR inhibitors. Validation studies using cytotoxicity assays confirmed our predictions that osimertinib shows a synergistic effect when combined with known KRAS G12C inhibitors (Section 3.7). Gene expression studies confirmed *KRAS* and mitogen-activated protein kinase (*MAPK*) are key targets for controlling prosurvival signals in KRAS G12C cells, as levels of expression of both were significantly altered following treatment with our predicted compounds [35].

## 2. Results and Discussion

### 2.1. Benchmarking and Drug Candidate Generation Using the CANDO Platform

CANDO v2.5 and its pipelines were applied as described in Figure 2. The average indication accuracy using CANDO v2.5 was ≈22–44% for the top10 through top100 cutoffs for 1650 indications with at least two approved drugs [21,31]. In contrast, the random control average values for these four cutoffs were ≈5–28% across the same indications. For NSCLC, with 47 approved drugs, the indication accuracy was ≈60–91% for the four cutoffs. Given that the indication accuracy for NSCLC was ≈60% at the top10 cutoff, over ten times the corresponding random control value of ≈5%, we expected our accuracy for generating mutation specific NSCLC putative candidates to be similarly high.

We first applied the CANDO platform to compare a subset of four KRAS inhibitors to our experimental compound/drug library, which identified EGFR inhibitors as being among the top most similar compounds. The literature confirmed this result, specifically showing effectiveness of second generation EGFR inhibitors against KRAS-mutant tumors. Therefore we pursued EGFR inhibitors as compounds of interest for targeting KRAS G12C NSCLC [6]. Compared to four other available EGFR inhibitors in our entire compound library, osimertinib was ranked as most similar to the four KRAS inhibitors overall, and ranked first in similarity to KRAS inhibitor BAY-293 (Figure 3). Taken together, these results suggested potential synergy between KRAS G12C inhibitors and EGFR inhibitors, particularly osimertinib, which led to our in vitro validation studies.

### 2.2. Validation of Osimertinib, ARS-1620, and BAY-293

Table 1 shows the two cell lines used for validation. NCI-H1792 used as the experimental cell line as it contains the KRAS G12C mutation, and PC-3 was used as the KRAS wild type/control cell line. H1792 was chosen specifically as it has exhibited resistance to ARS-1620 alone, and therefore is an ideal candidate for testing whether EGFR inhibitors in combination with ARS-1620 may mediate this resistance [35]. As predicted with the aid of CANDO, KRAS G12C inhibitor ARS-1620 combined with osimertinib and pan-KRAS inhibitor BAY-293 combined with osimertinib showed synergistic effects on decreasing cellular viability in H1792. Neither KRAS inhibitor was effective at decreasing cellular proliferation of H1792 alone, so GI50 values could not be determined (Appendix A). Figure 4 shows the GI50 for osimertinib with the H1792 cell line. We did not observe as strong effects on cellular proliferation with the above combinations of inhibitors when tested on the PC-3 cell line, indicating a decreased effect on KRAS wild type (Appendix A).There were no significant effects on PC-3 between the untreated control and the treatment conditions with the combination of ARS-1620 and osimertinib. However, the combination of BAY-293 and osimertinib did show a significant increase in inhibition for the highest three treatment conditions, albeit a much smaller increase compared to the effect observed with H1792. We believe this is due to BAY-293 targeting the interaction between WT-KRAS and SOS1, as well as mutant KRAS and SOS1. None of the other EGFR/ERbB inhibitors tested (erlotinib, gefitinib, afatinib, dacomitinib) showed synergy with ARS-1620, or effectiveness in decreasing H1792 cellular proliferation alone.

These findings are unexpected, especially since afatinib has been shown to demonstrate synergy with adagrasib/MRTX-849 in other studies using in vitro validations [2]. However, drug synergies may vary depending on which KRAS G12C inhibitor is used, and since our experiments were conducted with ARS-1620 this may account for the difference in results. Our research shows that osimertinib has the ability to decrease H1792 proliferation significantly alone as well as in combination with KRAS inhibitors ARS-1620 and BAY-293, respectively, (Figure 4, Figure 5 and Figure 6), as hypothesised by the CANDO platform. This indicates osimertinib may be more effective at inhibiting KRAS G12C NSCLC than other EGFR inhibitors such as afatinib, with greater capability to synergize with KRAS inhibitors [35].

### 2.3. Gene Expression Confirmation of Synergy

Figure 7 shows the results of the QPCR experiments to investigate the genetic components of the synergies we found using osimertinib with ARS-1620 and BAY-293. Previous QPCR data has been used similarly for combinations of PI3K inhibitors and KRAS G12C inhibitors, which have also shown synergistic capability in inhibiting cellular proliferation of KRAS G12C cell lines [35]. These results showed that MAPK reactivation is a key component of KRAS inhibitor drug resistance [35]. Here, we used increased DUSP-6 mRNA levels an indicator of MAPK activation as it has been well-established [36]. Suppression of KRAS mRNA was also evaluated to determine the effect on KRAS gene expression following direct inhibition of KRAS G12C [35].

Surprisingly, our QPCR results showed a significant increase in DUSP-6 expression following treatment with osimertinib alone, which may be due to compensatory mechanisms following protein inhibition. However, treatment with BAY-293 significantly decreased DUSP-6 expression levels compared to the control. Similarly, KRAS levels were significantly decreased relative to the control when treated with BAY-293 alone as well as BAY-293 and osimertinib combined. The combination of ARS-1620 and osimertinib, as well as ARS-1620 alone, was not significantly different compared to the untreated condition, for either gene of interest. The lack of KRAS inhibition with ARS-1620 and osimertinib is less surprising than the lack of DUSP-6 inhibition, since previous results also showed that KRAS expression did not predict response to treatment with ARS-1620 alone [35]. Since BAY-293 most significantly decreased levels of DUSP-6 mRNA, this suggests that BAY-293 works primarily through suppressing MAPK, while osimertinib and ARS-1620 inhibit KRAS cellular proliferation through inhibiting additional downstream mechanisms that are yet to be investigated.

Overall, our results provide strong evidence that osimertinib has the capability to synergize with BAY-293 and ARS-1620, even though it was designed to treat mutant NSCLC by specifically targeting the resistance conferring EGFR mutation T790M [37]. Viability assays confirmed that osimertinib synergizes with KRAS inhibitors, as predicted by the CANDO platform. Osimertinib has been shown to target ErbB2 [37], an oncoprotein that has been implicated in KRAS inhibitor resistance [11]; therefore it is possible that inhibition of ErbB2 may be responsible for the synergy of osimertinib with ARS-1620 and BAY-293.

### 2.4. Limitations and Future Directions

Drug/compound concentrations used were partially derived from previously identified GI50/IC50 values [5,35,37,38,39] but non-conformity in compound libraries, time points, and cell lines used makes direct comparisons difficult. Investigation with lower doses of compounds in the nM–pM range, along with other KRAS G12C mutant cell lines such as H358, is planned for future work as these concentrations are more relevant for use in patient studies. Additionally, experiments at the 72 h and 96 h time points with H1792 and other cell lines are planned.

Although our QPCR data showed the expected decrease of MAPK activation with the combination of osimertinib and BAY-293, we did not see the same trend for osimertinib and ARS-1620. More studies are needed to investigate the potential mechanisms of action for the synergy between osimertinib and ARS-1620. Specifically, ErbB2 silencing experiments with si-RNA are needed to examine the effects of ARS-1620 treatment on cells with ErbB2 knockdown, and other genes of interest such as SOS1. Altogether, our research shows the combination of osimertinib and KRAS inhibitors is a promising new potential treatment for KRAS G12C NSCLC, as well as KRAS G12C driven cancers in general.

## 3. Materials and Methods

### 3.1. CANDO Drug Discovery, Repurposing, and Design Platform

Figure 2 illustrates the precision medicine pipeline implemented within the CANDO platform for generating and validating putative NSCLC candidates. The CANDO platform for multiscale therapeutic discovery, repurposing, and design generates novel drug predictions and repurposes existing drugs for every indication by overcoming the limitations of traditional single target approaches [21,22,23,24,25,26,27,28,29,30,31,32,33,34]. One of the key tenets of CANDO is drugs that are safe for human use exert their therapeutic effects and undergo the process of absorption, dispersion, metabolism, and excretion (ADME) by interacting with multiple targets. Additionally, due to their size small molecule drugs are likely to engage in off-target interactions, i.e., have low energy of binding by chance to macromolecules not implicated in efficacy and/or ADME. CANDO exploits this inherent multitargeting nature of small molecules by computing interaction signatures that describe compound behaviour at multiple scales. Putative drug candidates are generated via a series of distinct pipelines that can be customized for the user’s individual needs, such as precision medicine in this case. The default pipeline first generates protein-compound interaction scores between every compound and every protein in their corresponding libraries using various algorithms for interaction scoring, such as those based on molecular docking and/or machine learning. Once the interaction scores have been calculated, drug candidates are generated by comparing the interaction signature of the query compound to that of every other compound. This comparison yields rankings based on the signature similarity between the ranked and query compound, with the hypothesis that higher ranked compounds are more likely to exhibit similar biological behavior to the query compound [21,22,23,24,25,26,27,28,29,30,31,32,33,34]. CANDO and its components have been extensively validated in laboratory studies for drug discovery, repurposing, and design [21,22,24,25,30,40,41,42,43,44,45,46,47,48]; however, this is the first instance of applying CANDO to precision oncology.

### 3.2. Compound and Protein Library Curation

Our compound library comprises 13,218 total drugs (2449 approved drugs and 10,769 investigational/experimental compounds), curated primarily using Drugbank [49]. Currently approved drugs and experimental compounds to treat KRAS G12C NSCLC, such as AMG-510/sotorasib, ARS-1620, MRTX-849/adagrasib, BAY-293 were manually added to our compound library to generate the most accurate novel compound and synergy predictions.

Three proteome libraries were used in this study: (1) 5317 human-specific proteins from the Protein Data Bank (PDB) [50] were curated to represent the “Human” proteome, as we are interested only in drugs capable of targeting KRAS G12C in humans. (2) The Cancer Genome Atlas’s (TCGA) study of lung adenocarcinoma was used to identify a list of eighteen genes that were mutated with statistical significance, based on an analysis of 230 lung adenocarcinoma patient samples [51]; the corresponding protein library was used to screen our initial predictions based on relevance to NSCLC, to increase our accuracy for novel drug predictions and repurposing of existing NSCLC treatments for KRAS mutations (“TCGA NSCLC”). (3) Additionally, the Kyoto Encyclopedia of Genes and Genomes (KEGG) [52,53,54] was used to identify genes both upstream and downstream of KRAS that are commonly affected by mutations in the oncogene. This yielded a further subset of nine genes whose corresponding proteins are key targets for KRAS-mutant NSCLC to further refine our predictions ( “KEGG”).

### 3.3. Compound-Protein Interaction Scoring Using a Bioanalytic Docking Approach

The bioanalytic docking (BANDOCK) compound-protein interaction scoring protocol was used to compute scores between every compound and every protein in the corresponding libraries [22,23,31]. BANDOCK uses the COACH algorithm to identify binding sites for each protein [31,55], which is based on the similarity of a given compound to known ligands in solved protein structures. For a given protein and ligand pair, COACH outputs a set of predicted ligands for each binding site, which BANDOCK then uses to compare to the compounds of interest via chemical fingerprinting, to assess for the presence or absence of shared chemical substructures using RDkit [56]. The maximum resulting Tanimoto coefficient (i.e., the strongest match) and its associated binding site score are then used to compute the final interaction score for the compound-protein pair, depending on the scoring protocol parameters [31]. For each compound, this process is repeated for every protein in the proteome library resulting in a vector of interaction scores known as the compound-proteome interaction signature.

### 3.4. Benchmarking

The benchmarking protocol in CANDO involves comparing the signatures of every approved compound against each other and ranking them by their similarity [21,22,23,24,25,26,27,28,29,30,31,32,33,34]. These rankings are then assessed to see if the drugs approved to treat a particular indication are observed within specific cutoffs. This protocol is run iteratively for every indication (2257 in v2+) with at least two approved drugs (1650), and the resulting accuracy scores are averaged across all indications. We use numerous metrics to assess performance [21], but a primary intuitive one is the indication accuracy which is calculated based on the number of times an approved drug for a given indication is observed within various cutoffs, such as top10, top25, top50, and top100. This indication accuracy is calculated using the equation *c*/*d* × 100, where *c* is the number of times at least one other drug approved for the same indication was recovered for a given cutoff, and *d* is the total number of drugs approved for that indication [21,31]. This process is repeated for every indication in our indication library and averaged to obtain an overall average indication accuracy.

### 3.5. Putative Drug Candidates

To generate our putative drug candidates, pairwise similarity scores were calculated between the compound-proteome interaction signatures of known KRAS inhibitors and all other compound-proteome signatures. These candidates were then ranked by similarity of their interaction signatures, calculated using the cosine distance between the two vectors representing the signatures. Compounds with more similar signatures are hypothesized to exhibit more biologically similar behavior. We therefore used these similarity scores to known NSCLC drugs such as AMG-510/sotorasib and MRTX-849/adagrasib, in addition to experimental compounds such as BAY-293 and ARS-1620, to identify candidates that may be capable of treating KRAS G12C NSCLC in addition to their intended use. We also identified other drugs/compounds in our library that may be repurposed for treating KRAS G12C NSCLC. The most promising drug/compound predictions were further investigated via in vitro validation studies.

### 3.6. Cell Lines and Compounds Chosen for Validation

Human cancer cell lines NCI-H1972 (CRL-5908™, non-small cell lung cancer adenocarcinoma), and PC-3 (CRL-1435™, a grade IV prostatic adenocarcinoma) were obtained from American Type Culture Collection and were grown in F-12K Medium (Kaighn’s Modification of Ham’s F-12 Medium) and RPMI 1640 media, respectively, supplemented with 10% fetal bovine serum, 100 μmL penicillin, and 100 μg/mL streptomycin. Cultures were maintained at 37 °C in a humidified incubator with 5% CO_2_. KRAS inhibitors ARS-1620, BAY-293, and EGFR inhibitors erlotinib, gefitinib, afatinib, dacomitinib, and osimertinib were purchased from Selleck Chemicals (based on the predictions made in Section 3). Drug/compound concentrations used were partially based on reported values from the literature [5,35,37,38,39].

### 3.7. Cytotoxicity Assays

We examined the effects of ARS-1620 and BAY-293 as well as the EGFR inhibitors erlotinib, gefitinib, afatinib, dacomitinib, and osimertinib on cell viability by performing dose and time kinetic studies using the CCK-8 assay (Cat #CK04 Dojindo Molecular Technologies). The CCK-8 assay uses colorimetric detection for the determination of cell viability in cytotoxicity assays. The water-soluble tetrazolium salt, WST-8, is reduced by dehydrogenase activities in cells to produce a yellow-color formazan dye, which is soluble in the tissue culture media. The amount of the formazan dye, generated by the activity of dehydrogenases in cells, is directly proportional to the number of living cells. Briefly, cells were seeded at a density of 10,000 cells/100 μL/well in 96 well plates in complete growth medium. Cells were incubated with the desired varying drug concentration for 24–48 h. No drug (untreated cells) was used as the control. At the end of the incubation period, media was then replaced with 100 μL phenol-red free media and 10 μL of CCK-8 solution followed by a two hour incubation. The absorbance was then read at 450 nm using a spectrophotometer (Biotek, Winooski, VT, USA) plate reader. All drug concentrations were tested in triplicate.

### 3.8. Gene Expression Assays by QPCR

The effects of ARS-1620, BAY-293, and osimertinib on KRAS and the dual specificity phosphatase-6 (DUSP-6) gene expression in NCI-H1792 cells were examined by RT-qPCR to determine the genetic component of our viability results. Total RNA was extracted from the tumor cell lines using the TRIzol™ reagent (Invitrogen, Carlsbad, CA, USA), as per protocol provided by manufacturer. RNA quantity and purity were assessed via a 260/280 nm ratio using a NanoDrop2000™ (Thermo Fisher Scientific, Wilmington, DE, USA). 500 ng of total RNA was reverse transcribed using the All in One cDNA synthesis Supermix™ (Bimake.com) to obtain cDNA for QPCR. QPCR was performed using the AzuraView GreenFast qPCR blue Mix LR™, a ready to use PCR SYBR green master mix of fluorescent dye for measuring double stranded DNA (Azura Genomics Inc.). QPCR was carried out using the following conditions: 95 °C for 3 min, followed by 40 cycles of 95 °C for 40 s, 60 °C for 30 s, and 72 °C for 1 min; the final extension was at 72 °C for 5 min. Validated primers were obtained from Integrated DNA technologies and the final primer concentration used in the PCR was 0.1 μM. The primer sequences of the DUSP6 and KRAS2 gene were as follows: DUSP-6 forward primer: 5′-CGACTGGAACGAGAATACGG-3′, DUSP-6 reverse primer: 5′-TTGGAACTTACTGAAGCCACCT-3′; KRAS2 forward primer: 5′-TGTTCACAAAGGTTTTGTCTCC-3′, KRAS2 reverse primer: 5′-CCTTATAATAGTTTCCA TTGCCTTG-3′. The housekeeping gene β-Actin was used as the control. Relative gene expression of DUSP6, KRAS, β-actin mRNA in control versus drug treated samples were assessed in NCI-H1792 cells. Gene expression was calculated using the comparative threshold cycle (Ct) method. After the Ct of each sample was determined, the relative level of a transcript (2ΔCt) was calculated by obtaining ΔCt (test Ct −β-Actin Ct), and the transcription accumulation index ΔΔCt = (CtTarget − Ctβ-actin)${}_{treated}$ − (CtTarget − Ctβ-actin)${}_{untreated}$ [57].

### 3.9. Statistical Analysis

All experimental data were analyzed using Graph.Pad Prism 9.0 (GraphPad Software La Jolla, CA, USA). Results are expressed as mean ± standard deviation (SD). The student *t*-test was used for comparisons between control and treated groups. Statistical significance was set at *p* value < 0.05. The drug concentration (μM) required to inhibit 50% net of cell growth (growth inhibition at 50% or GI50) was calculated using GraphPad Prism Software version 9.0 using the functions log(inhibitor) or inhibitor depending on whether a log-scale dilution was used, versus a response-variable slope (four parameters). Values were capped at the maximum drug dose used. Synergy was quantified using CompuSyn Software demonstrating at least a two fold shift in potency between single and combination drug doses [58].

## 4. Conclusions

Our research demonstrates a novel application of the CANDO platform for identifying putative drug candidates targeting specific oncogenic mutations. Here we show that even drugs designed to target specific protein mutations, such as osimertinib, are capable of targeting multiple proteins, which has been corroborated in the literature [12,37]. As shown previously, our platform succeeded in predicting synergies between osimertinib and ARS-1620 as well as osimertinib and BAY-293 using a multiscale approach. Therefore, our precision medicine pipeline within CANDO may be used in the future to predict both existing drugs and novel compounds capable of targeting single nucleotide variations, and supplement traditional high-throughput and fragment-based experiments. This approach allows for additional safety measures for patients by utilizing existing drugs for new uses, speeds the development of treatments for particularly aggressive oncogenic driver mutations lacking viable treatment options, and adds valuable proteome and other multiscale information to existing drug discovery methods. Validation studies for the top candidates from our full drug/compound library are in progress to determine if other compounds synergize with available KRAS G12C inhibitors, or achieve inhibition alone. Novel designs based on mutant-specific inhibitors have also been synthesized and are also in the process of being validated [59]. The goal of this study is to develop sophisticated multiscale hybrid computational/experimental pipelines for precision drug discovery that are capable of rapidly generating safe and effective putative drug candidate leads, along with elucidating mechanistic details that may be used as a basis for further hypothesis generation. To that end, this study validated effective drug combinations generated using the CANDO platform for multiscale therapeutic discovery to target driver KRAS mutations in NSCLC.

## Figures and Tables

**Figure 1 ijms-24-00997-f001:**
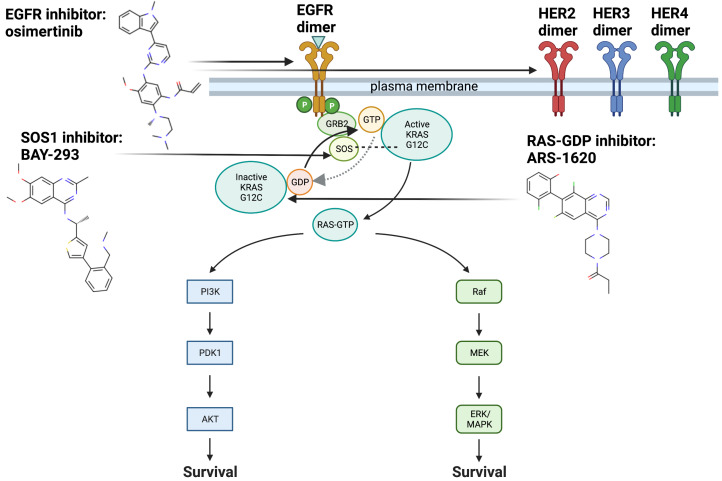
EGFR, SOS1, RAS and downstream pathways with drugs of interest and mechanisms of action. KRAS G12C undergoes biochemical changes that allow it to remain in its GTP-bound active state for longer periods of time, versus the inactive GDP-bound state: Increased time in the GTP-bound active state leads to increased activation of downstream signaling cascades involving Raf-MEK-ERK and PI3K-AKT-mTOR which are involved in cellular proliferation and survival [1]. Here we show the targets for inhibitors analyzed in this study, BAY-293, ARS-1620, osimertinib, and relevant downstream signaling pathways. BAY-293 targets the KRAS G12C-SOS1 interaction, which decreases the exchange of GDP to GTP involved in activating KRAS G12C. ARS-1620 targets GDP-bound KRAS G12C, and decreases the ability of SOS1 to exchange GDP to GTP. Osimertinib, predicted by our computational platform during the course of this study, acts upstream of KRAS G12C to inhibit EGFR, specifically the resitance-causing T790M mutation. It has also demonstrated activity against wild type EGFR and HER2, which have been implicated in adaptive resistance to KRAS G12C inhibition through re-activation of scaffold proteins. Although these three inhibitors (ARS-1620, BAY-293, osimertinib) have differing methods of action, it is possible they may synergize through shared targets that are as yet unknown. Created with BioRender.com.

**Figure 2 ijms-24-00997-f002:**
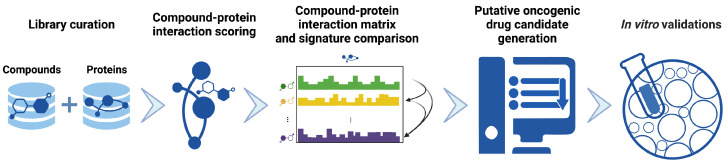
Precision drug discovery pipeline within CANDO for generating novel drug leads for KRAS G12C NSCLC. We created a pipeline within the previously developed CANDO platform [21,22,23,24,25,26,27,28,29,30,31,32,33,34] to generate drug candidates based on novel applications of existing drugs/compounds using four approved or investigational treatments for KRAS G12C NSCLC (AMG-510/sotorasib, ARS-1620, MRTX-849/adagrasib, BAY-293). Our hypothesis is that these drugs/compounds may be used to identify other behaviorally similar drugs/compounds that could (also) be effective against KRAS G12C NSCLC, either individually or synergistically. Interaction scores between every compound in our 13,218 drug/compound library and every protein in our 5317 human proteome library were first calculated using bioanalytical docking. This resulted in a compound-proteome interaction signature for each drug/compound, i.e., a vector of 5317 scores that describes the functional behaviour of the compound, shown here with proteins depicted horizontally, and compounds vertically as an interaction matrix (Section 3.3). Compound-compound behavioral similarity was then calculated by comparing individual compound-proteome interaction signatures using the cosine distance and benchmarked with known data to ensure optimal performance (Section 3.4). Putative drug candidates were then generated from the drugs/compounds with the most similar pairwise interaction signatures to those belonging to investigational and approved treatments of KRAS G12 NSCLC (Section 3.5). Signature comparison results indicated that osimertinib was most similar to KRAS inhibitor BAY-293, and highly similar to KRAS G12C inhibitor ARS-1620 and other KRAS inhibitors. Finally, these candidates were used either individually or in combination, for validation in human cancer cell lines (Section 1 and Section 3.7). Created with BioRender.com.

**Figure 3 ijms-24-00997-f003:**
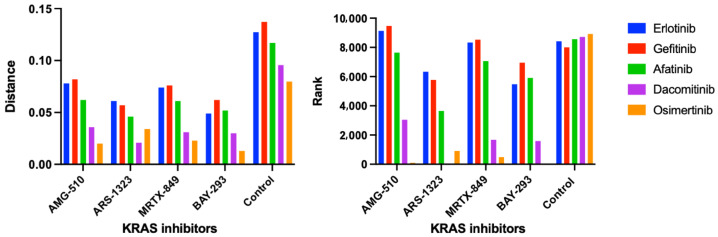
KRAS and EGFR inhibitor similarity scores and rankings using the CANDO platform. Following the generation of the compound-proteome interaction signatures for all drugs/compounds in our library, the similarity scores and rankings were calculated between four KRAS inhibitors AMG-510/sotorasib, ARS-1323 (racemate of ARS-1620), MRTX-849/adagrasib, BAY-293, and all other compound signatures. Overall, we found high similarity between our four KRAS inhibitors and EGFR inhibitors erltotinib, gefitinib, afatinib, dacomitinib, and osimertinib using the cosine distance between the interaction signature vectors (Section 3.5). The left panel shows the average similarity scores (vertical axis) for the five first, second, and third generation EGFR inhibitor interaction signatures (colored bars) compared to the signatures of four KRAS inhibitors (horizontal axis). The average similarity scores for each EGFR inhibitor signature compared to those from our entire 13,217 drug/compound library are also shown as a control. The corresponding pairwise signature similarity rankings (vertical axis) are displayed on the right panel. Overall, KRAS G12 inhibitors were predicted to be highly similar to EGFR inhibitors relative to the average drug/compound in our library. Osimertinib, a third generation EGFR inhibitor, was calculated to be the most similar to all the four KRAS inhibitors relative to the other EGFR inhibitors, and was ranked first when compared to BAY-293 relative to the entire compound library. Created with BioRender.com.

**Figure 4 ijms-24-00997-f004:**
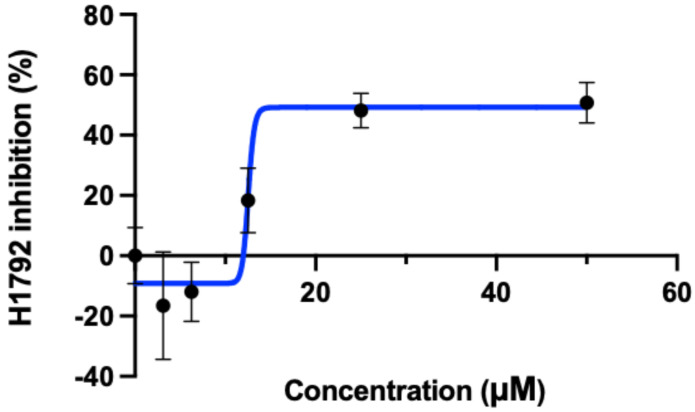
GI50 for osimertinib. The concentration (horizontal axis) of osimertinb is plotted against the H1792 cellular inhibition percentage (vertical axis). The concentration required for 50% cellular inhibition (GI50) using osimertinib was calculated to be 12.54 μM using Graphpad prism 9.0. Although osimertinib was designed to target EGFR-mutant cell lines, here we show that it is also capable of inhibiting KRAS G12C cells as inferred from the predictions made by the CANDO platform. Created with BioRender.com.

**Figure 5 ijms-24-00997-f005:**
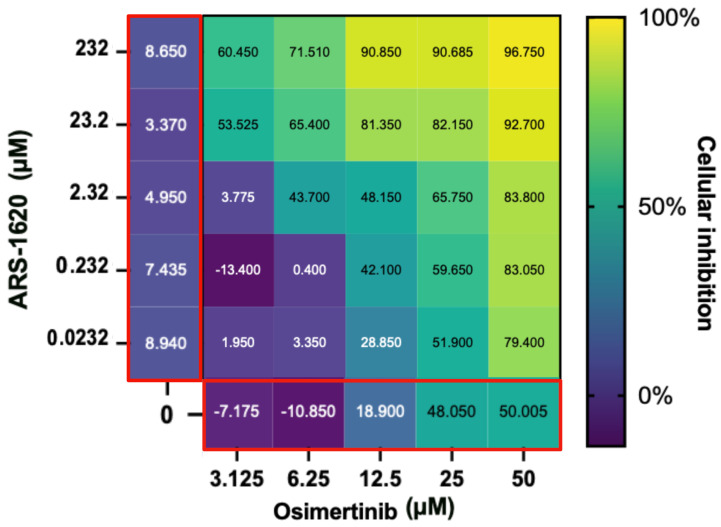
Evaluation of the ARS-1620 and osimertinib drug combination. The main heatmap panel outlined in black shows H1792 cell inhibition as a percentage normalized using untreated controls ranging from 0% (dark blue) to 100% (yellow) for different concentrations. ARS-1620 and osimertinib by themselves are shown on the left and the bottom, respectively, (both outlined in red). This shows that although H1792 did not respond strongly to ARS-1620 alone, a synergistic effect on cellular inhibition was observed when combined with osimertinib as hypothesised from the CANDO predictions. Created with BioRender.com.

**Figure 6 ijms-24-00997-f006:**
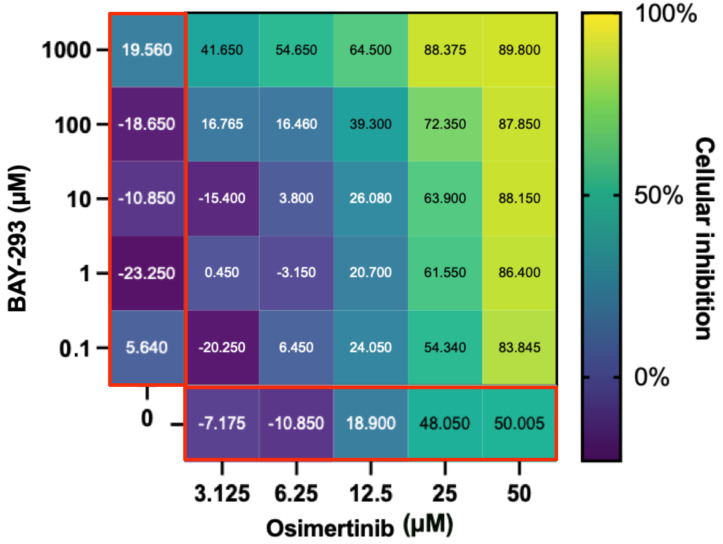
Evaluation of the BAY-293 and osimertinib drug combination. The main heatmap panel outlined in black shows H1792 inhibition as a percentage normalized using untreated controls ranging from 0% (dark blue) to 100% (yellow) for different concentrations. BAY-293 and osimertinib by themselves are on the left and bottom, respectively, (both outlined in red). This shows that although H1792 did not respond strongly to BAY-293 alone, a synergistic effect on cellular inhibition was observed when combined with osimertinib. Created with BioRender.com.

**Figure 7 ijms-24-00997-f007:**
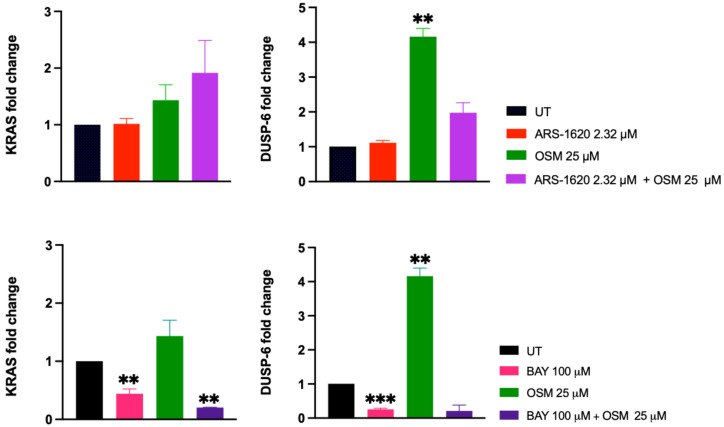
Gene expression analysis of synergistic drug combinations. The top two panels show KRAS and DUSP-6 expression levels upon treatment with ARS-1620, osimertinib, and combination at 24 h; similarly, the bottom two panels show KRAS and DUSP-6 expression levels upon treatment with BAY-293, osimertinib, and combination at 24 h. Treatment with osimertinib significantly increased DUSP-6 expression relative to the untreated control (*p* < 0.01; **). There were no significant differences between any of the treatment conditions of KRAS expression levels for the ARS-1620 and osimertinib conditions. However, treatment with BAY-293 significantly decreased expression of DUSP-6 compared to the untreated control (*p* < 0.001; ***). Treatment with BAY-293, and BAY-293 combined with osimertinib, also significantly decreased the expression of KRAS compared to the untreated control, albeit to a lesser extent (*p* < 0.01). These findings indicate BAY-293 inhibits H1792 proliferation through suppression of MAPK as measured by decreased DUSP-6, while osimertinib and ARS-1620 inhibit proliferation through mechanisms that need to be further elucidated. Created with BioRender.com.

**Table 1 ijms-24-00997-t001:** Cell lines used for validation. The name of the cell line, tumor type, and the mutation status are shown. NCI-H1792 was used for the experimental cell line, since it contains the target KRAS G12C mutation, and PC-3 was used as the control cell line since it contains wild type KRAS.

Cell Lines	Histology	KRAS Exon 2 Mutation Status
NCI-H1792	Adenocarcinoma	KRAS G12C, homozygous
PC-3	Adenocarcinoma	KRAS wild type

## Data Availability

Data including compound-protein interaction matrices are available upon request.

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
