# Peer review of "Multiscale Analysis and Validation of Effective Drug Combinations Targeting Driver KRAS Mutations in Non-Small Cell Lung Cancer"

_ijms, 2023, doi:10.3390/ijms24020997_

Round 1
Reviewer 1 Report
The authors screened and determined a potential inhibitor that exerts a synergistic effect with KRASG12C inhibitor using the Computational Analysis of Novel Drug Opportunities (CANDO) platform. Osimertinib in combination with the KRASG12C inhibitor ARS-1620 and the pan-KRAS inhibitor BAY-293 synergistically decreased cellular proliferation. This effect was not observed for other EGFR-TKIs, such as gefitinib, erlotinib, afatinib, and dacomitinib.
Despite these noteworthy observations, some points remain unaccounted for in this model. Hence, there is insufficient information to warrant the publication of this manuscript in its present form.
Comments:
Lines 93-94: The cited reference is inappropriate. Reference 11 does not indicate that osimertinib exhibits increased ability to target ErbB2/HER2 compared to second generation inhibitors.
Figure 3 is difficult to understand and needs to be revised for improved clarity.
Line 314: The authors indicated that H1792 cells were resistant to ARS-1620 alone and cited reference 34. In the cited paper, 10 mM ARS-1620 inhibited cell proliferation of H1790 cells. However, Figure 5 indicates that ARS-1620 did not inhibit cell proliferation even at 230 mM. Please explain this discrepancy.
The concentrations of inhibitors used in this article, such as osimertinib, ARS-1620, and BAY-293, were too high to permit specific inhibition. These concentrations would produce a non-specific effect. The authors should perform these assays again using concentrations that are relevant in humans.
Author Response
We would like to thank the reviewers and editors for their time, consideration, and thoughtful critiques. Below are our responses to the reviewer feedback.
Reviewer 1 -
``Lines 93-94: The cited reference is inappropriate. Reference 11 does not indicate that osimertinib exhibits increased ability to target ErbB2/HER2 compared to second generation inhibitors. ''
The authors of reference 11 reviewed various papers examining the effectiveness of osimertinib vs other EGFR inhibitors and pan-ErbB inhibitors for targeting HER2. In particular they mention Liu et. al ( 2018) as finding increased effectiveness of osimertinib for treating mouse models of HER2 overexpressing NSCLC, compared to erlotinib and afatinib, first and second generation EGFR inhibitors, respectively. However, for mouse models of HER2 exon 20 mutant NSCLC afatinib was more effective than osimertinib, indicating that osimertinib is only more effective than second generation EGFR inhibitors in specific contexts. Additionally, Ho et. al (2021) found that HER2 overexpression models of NSCLC were resistant to treatment with sotorasib/ AMG-510 alone, but responded when co-treated with SHP2 inhibitors. These clarifications have been added to the manuscript.
``Figure 3 is difficult to understand and needs to be revised for improved clarity. ''
The figure caption has been adjusted to more clearly describe the results.
``Line 314: The authors indicated that H1792 cells were resistant to ARS-1620 alone and cited reference 34. In the cited paper, 10 mM ARS-1620 inhibited cell proliferation of H1790 cells. However, Figure 5 indicates that ARS-1620 did not inhibit cell proliferation even at 230 mM. Please explain this discrepancy. ''
For reference 34, although Misale et. al (2019) did see a decrease in proliferation at 10 micromolar ARS-1620, they completed the assay at the 96 hour time point while we only used a 48 hour time point for all cellular proliferation assays, which likely accounts for the difference.
``The concentrations of inhibitors used in this article, such as osimertinib, ARS-1620, and BAY-293, were too high to permit specific inhibition. These concentrations would produce a non-specific effect. The authors should perform these assays again using concentrations that are relevant in humans.''
Figure 4 contained an error in the horizontal axis label, millimolar was used instead of micromolar. This has been corrected, and all figures now show the correct units. We apologize for any confusion caused by this error, however our actual concentrations were in the nanomolar to millimolar range which encompasses concentrations relevant to humans, and is similar to concentrations used by Ho et. al (2021) and Misale(2019), particularly as these concentrations were further diluted by a factor of ten once added.

Reviewer 2 Report
Reviewed manuscript aimed to study the drug combinations targeting KRAS mutations by the CANDO platform. Data were verified by in vitro studies.
The article might be interesting for readers but needs major text revision. Here, I present my major comments.
1. The title and abstract did not reflect the content of the whole article. The authors should highlight the novelty of the studies (like that they first applied the CANDO platform to compare a subset of KRAS inhibitors), rather than focus on achievings from the previous article. Furthermore, should add information that cytotoxicity and gene expression by qPCR studies were done.
2. Definitely, in the introduction, the authors should add a precisely described goal of study and information on how they design the methods to achieve that.
3. I have a feeling that in the Materials and methods section authors are repeating data about the CANDO platform from the introduction. I wondering what is new in this section. These two sections (Introduction and Materials and methods) should be reviewed in that context.
4. I understand that the PC-3 cell line was used as the control because contains wild-type KRAS, but why authors did not use the normal cell line? This should be clarified.
5. There is no clear, separate discussion section. Therefore, in other sections, especially Results, the authors should discuss obtained data.
Author Response
We would like to thank the reviewers and editors for their time, consideration, and thoughtful critiques. Below are our responses to the reviewer feedback.
Reviewer 2 -
``Reviewed manuscript aimed to study the drug combinations targeting KRAS mutations by the CANDO platform. Data were verified by in vitro studies.
The article might be interesting for readers but needs major text revision. Here, I present my major comments.
1. The title and abstract did not reflect the content of the whole article. The authors should highlight the novelty of the studies (like that they first applied the CANDO platform to compare a subset of KRAS inhibitors), rather than focus on achievings from the previous article. Furthermore, should add information that cytotoxicity and gene expression by qPCR studies were done.''
The title has been adjusted to reflect the use of CANDO for our putative drug candidates. Likewise, the abstract has been adjusted accordingly.
``Definitely, in the introduction, the authors should add a precisely described goal of study and information on how they design the methods to achieve that. ''
We have added a goal statement to the end of the introduction.
``I have a feeling that in the Materials and methods section authors are repeating data about the CANDO platform from the introduction. I wondering what is new in this section. These two sections (Introduction and Materials and methods) should be reviewed in that context.''
Repetitive verbiage mentioning CANDO in the introduction has been removed. The information regarding CANDO in the introduction is more broad, versus the information in the methods goes into specific detail as to the components of the pipeline.
``I understand that the PC-3 cell line was used as the control because contains wild-type KRAS, but why authors did not use the normal cell line? This should be clarified. ''
The PC-3 cell line was used in this case since we were mainly interested in validating our predictions, and therefore establishing they were only effective for KRAS mutant, not WT cells. We thank the reviewer for this suggestion, and for future studies that aim to clarify the mechanisms our findings and explore safety in humans we will use a non-cancerous lung cell line for comparison.
``There is no clear, separate discussion section. Therefore, in other sections, especially Results, the authors should discuss obtained data.''
The results are discussed in each of the corresponding individual subsections under the combined results and discussion section. We have added additional information in section 4.2 discussing figure 4 and the supplementary figures so that every figure is described in detail in its corresponding section.

Reviewer 3 Report
This manuscript demonstrates use of a computational pipeline for precision drug discovery. More specifically, the manuscript uses the CANDO platform to identify drug combination mutant phenotypes of non-small cell lung cancer. The authors compared similarity scores between compound-proteome interaction signatures of KRAS inhibiters and all other compounds to obtain a list of compounds hypothesized to exhibit similar biological behavior. Their analysis suggests an EGFR inhibitor osimertinib is likely to have synergized effect with the four KRAS inhibiters analyzed in this study. The authors further validate their computational findings with initial cell proliferation assays. These results are interesting and encouraging, though more experimental investigation is needed to further understand the interaction mechanism of these compounds and their safety as pointed out by the authors. The paper is well written, the problem is clearly stated and well defined. The results are presented in a clear and logical way and backed by computational analysis and experiments and the conclusions are supported by the results presented. I recommend this paper for publication.
Author Response
We would like to thank the reviewers and editors for their time, consideration, and thoughtful critiques. Below are our responses to the reviewer feedback.
``This manuscript demonstrates use of a computational pipeline for precision drug discovery. More specifically, the manuscript uses the CANDO platform to identify drug combination mutant phenotypes of non-small cell lung cancer. The authors compared similarity scores between compound-proteome interaction signatures of KRAS inhibiters and all other compounds to obtain a list of compounds hypothesized to exhibit similar biological behavior. Their analysis suggests an EGFR inhibitor osimertinib is likely to have synergized effect with the four KRAS inhibiters analyzed in this study. The authors further validate their computational findings with initial cell proliferation assays. These results are interesting and encouraging, though more experimental investigation is needed to further understand the interaction mechanism of these compounds and their safety as pointed out by the authors. The paper is well written, the problem is clearly stated and well defined. The results are presented in a clear and logical way and backed by computational analysis and experiments and the conclusions are supported by the results presented. I recommend this paper for publication. ''
We thank the reviewer for their positive and thoughtful response.

Round 2
Reviewer 1 Report
The cited reference # 12 is inappropriate. Please cite original article.
The concentrations of inhibitors used in this article, such as osimertinib, ARS-1620, and BAY-293, were too high to permit specific inhibition.It would not similar to the concentrations used by Ho et. al (2021) and Misale(2019). The authors should mention the limitation regarding this matter in discussion section.
Author Response
We thank the reviewer for their thoughtful comments, please see the following responses below.
The cited reference # 12 is inappropriate. Please cite original article.- changed to original article (Liu et al. 2018).
The concentrations of inhibitors used in this article, such as osimertinib, ARS-1620, and BAY-293, were too high to permit specific inhibition. It would not similar to the concentrations used by Ho et. al (2021) and Misale(2019). The authors should mention the limitation regarding this matter in discussion section. -The manuscript has been adjusted accordingly in the manuscript in sections 3.6 “Cell lines and compounds chosen for validation”, and “Limitations and future directions”.
Section 3.6- Drug/compound concentrations used were partially based on reported values from the literature. Limitations and future directions- As osimertinib has a known ability to target ErbB2, which has been implicated in KRAS inhibitor resistance, it is possible inhibition of ErbB2 is responsible for the synergy between ARS-1620 and osimertinib (Liu2018). Drug/compound concentrations used were partially derived from previously identified GI50/IC50 values (Hillig2019,Janes2018,Cross2014,Misale2019,Liu2018) but non-conformity in compound libraries, time points, and cell lines used makes direct comparisons difficult. Investigation with lower doses of compounds in the nM---pM range, along with other KRAS G12C mutant cell lines such as H358, are planned for future work as these concentrations are more relevant for use in human patients. Additionally, future experiments at the 72 hour and 96 hour time points with H1792 and other cell lines are planned.
Reviewer 2 Report
The authors responded to all my comments and corrected the manuscript.
Author Response
We thank the reviewer for their time and consideration.
Round 3
Reviewer 1 Report
I do not have any further inquiries.